# Spherical SO(3) Equivariant Local Attention

Yusuke Sekikawa [* 1]   Jun Nagata [* 1]   Itsumi Araki [1]   Ruka Eto [1]

## Abstract

Spherical signals provide a natural representation for omnidirectional perception and often benefit from equivariance to 3D rotations. Recent spherical vision transformers implement local self-attention on spherical grids, but most retain only partial SO(3) equivariance and rely on *location-dependent* local positional embeddings (LPEs). Such LPEs can degrade robustness to camera tilt or object reorientation and introduce additional memory and computational overhead. We propose *Spherical* SO(3)-*Equivariant Local Attention* (𝕊*oLA*), an LPE-free local attention mechanism for spherical signals. 𝕊*oLA* achieves full SO(3) equivariance through a distance-preserving positional modulation that couples query/key features with each tokens unit direction. Specifically, the modulation lifts queries and keys using an outer-product with the 4D direction dependent vector. The induced similarity of the modulated queries and keys depends on content affinity and great-circle distance while remaining invariant to global SO(3) rotations. The same formulation admits a softmax-free linear variant that computes local attention via key-value aggregation without per-query neighbor materialization. We integrate 𝕊*oLA* into a U-shaped spherical transformer for 360° depth estimation and semantic segmentation, demonstrating substantially improved robustness to arbitrary 3D rotations compared to prior spherical transformers with similar computational costs.

## 1. Introduction

Spherical data are a natural representation for omnidirectional perception: they unify various camera models (e.g., perspective and fisheye) and provide a common domain for multi-camera fusion. This is particularly advantageous in

[1]DENSO IT Lab., Tokyo, Japan. Correspondence to: Yusuke Sekikawa <sekikawa.yusuke@core.d-itlab.co.jp>.

*Proceedings of the 43rd International Conference on Machine Learning*, Seoul, South Korea. PMLR 306, 2026. Copyright 2026 by the author(s).

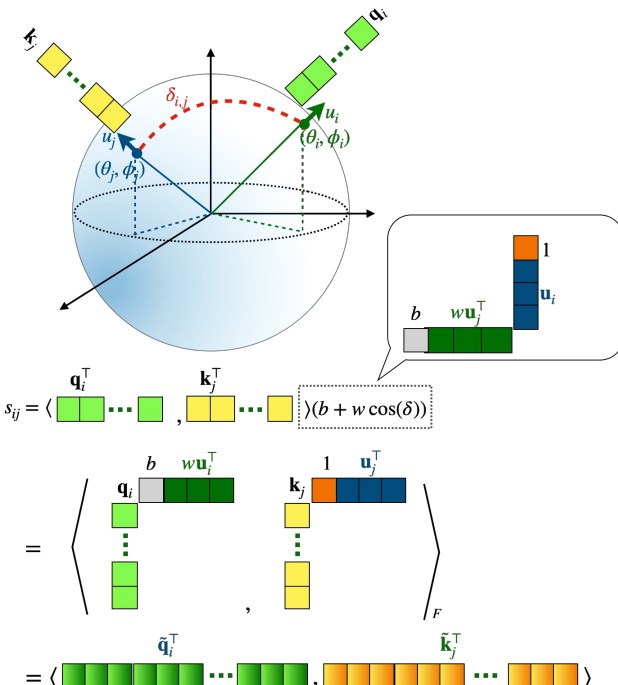

*Figure 1.* **Core mechanism of 𝕊*oLA*.** For each token direction $\mathbf{u}(\theta, \phi) \in \mathbb{S}^2$, we apply a separate direction-dependent outer-product modulation to query/key features. The resulting dot product factors into a content term $\langle \mathbf{q}_i, \mathbf{k}_j \rangle$ and a geometric term $b + w \cos \delta_{ij}$, where $\delta_{ij}$ is the great-circle distance. This embedding yields an SO(3)-equivariant similarity kernel without using location-dependent local positional embedding.

a variety of applications, including robotics, autonomous driving, and medical imaging. Many downstream tasks involving spherical signals require or benefit from SO(3)-equivariance. In the context of understanding indoor or outdoor scenes, the same object may appear at different latitudes or orientations. However, without rotation equivariance, the representation can vary solely based on the object's vertical position. In contexts such as astrophotography, meteorology, microscopic imaging, or factory automation (e.g., bin-picking), the direction of gravity can be irrelevant. In these cases, we ideally seek models that respond consistently when the entire spherical signal is globally rotated; i.e., models that are equivariant to the SO(3)-action on $\mathbb{S}^2$. Processing spherical signals under this symmetry is challenging because no *planar* parameterization preserves spherical geometry; thus, architectures designed for planar

images are difficult to apply directly.

A common workaround for processing spherical signals involves projecting spherical data onto one or more 2D gridssuch as equirectangular grids or collections of perspective grids (e.g., cubemaps)and then applying conventional regular grid models, such as CNNs or vision transformers. While intuitive and compatible with established planar backbones, these projection-based approaches are not rotation-equivariant and introduce position-dependent distortions and discontinuities, particularly near projection boundaries or view transitions (Tateno et al., 2018).

An alternative line of work processes spherical data natively via spherical CNNs that operate on $\mathbb{S}^2$ (Cohen et al., 2018; Esteves et al., 2018; Ocampo et al., 2023). These methods provide principled handling for groups such as SO(3); however, they often rely on spectral transforms, group operations, or irregular meshes, which are inherently computationally demanding and therefore difficult to scale, limiting their practical applications.

Recently, transformer-based approaches have attracted increased interest, aiming to combine the flexibility of attention with spherical geometry (Carlsson et al., 2024; Anonymous, 2025; Bonev et al., 2025). For example, SphereUFormer (Benny & Wolf, 2025) defines spherical local attention and incorporates it within a U-shaped architecture for 360° perception. While these methods achieve strong empirical performance, there remains significant room for improvement in both robustness and computational efficiency. Regarding robustness, many spherical transformers are equivariant only to a subgroup of SO(3); for instance, SphereUFormer is equivariant only to horizontal rotations, therefore, camera tilt or vertical motion can degrade its accuracy. In terms of computation, many spherical local attention mechanisms rely on location-dependent operations, such as local positional embeddings (LPEs) or spherical integration operations. These complex local operations introduce substantial latency overhead due to irregular memory access.

**Our approach.** We propose *Spherical* SO(3)-*Equivariant Local Attention* (SoLA, Figure 1), a LPE-free SO(3)-equivariant local attention mechanism that operates directly on spherical signals. The key idea for realizing this novel property is a *distance-preserving positional modulation* that couples content features with each token's direction. Concretely, for each feature location $i, j$, we form an outer-product modulation vector, $[b, w\mathbf{u}_i^\top]$ for queries and $[1, \mathbf{u}_j^\top]$ for keys, where $\mathbf{u} \in \mathbb{S}^2$ denotes the direction associated with each input token. For a query token $i$ and a key token $j$, we modulate as $\tilde{\mathbf{q}}_i = \mathbf{q}_i[b, w\mathbf{u}_i^\top]$ and $\tilde{\mathbf{k}}_j = \mathbf{k}_j[1, \mathbf{u}_j^\top]$ using outer-product lifting. The resulting query-key similarity $s_{ij}$, calculated using the Frobenius dot-product, factors into

content similarity and great-circle distance $\delta_{ij}$ as follows:

$$s_{ij} = \left\langle \tilde{\mathbf{q}}_i, \tilde{\mathbf{k}}_j \right\rangle_F = \langle \mathbf{q}_i, \mathbf{k}_j \rangle (b + w \cos \delta_{ij}),$$

from the relation of $\mathbf{u}_i^\top \mathbf{u}_j = \cos \delta_{ij}$. Because the geometric factor depends only on relative directions, it is invariant to global SO(3) rotations. Geodesic neighborhoods move together through global rotation; therefore, the attention mechanism based on the embedding yields an SO(3)-equivariant local attention operator without location-dependent LPE operations (See Section 3 for details).

When used in a *linear* (softmax-free) attention formulation, the order of multiplication can be rearranged such that the local key-value outer product aggregation is computed first, followed by the multiplication with each query. This formulation avoids the per-query materialization of neighbor features and enhances memory efficiency, particularly for local attention with large receptive fields or global attention.

We integrate SoLA into a U-shaped spherical transformer and evaluate it on 360° depth estimation and semantic segmentation benchmarks on Stanford2D3D (Armeni et al., 2017) and Structured3D (Zheng et al., 2019), including their globally rotated variants. We also demonstrate effectiveness in digit classification on the sphere and in forecasting the dynamics of the shallow water equations on the sphere, both of which strongly benefit from SO(3)-equivariance. Across these settings, SoLA demonstrates robust performance against rotations. Notably, due to its LPE-free formulation, SoLA achieves SO(3)-equivariance with a computational cost comparable to prior spherical transformers with limited equivariance.

## 2. Related work

### 2.1. Projection-based methods

Early methods for omnidirectional vision employed projection onto one or more 2D regular grid parameterizations–such as equirectangular projection (ERP) or collections of perspective projections (e.g., cubemaps)–and subsequently applied conventional 2D CNNs or vision transformers. Several studies have attempted to compensate for distortions by using specialized filters or sampling schemes. Distortion-aware convolutions adapt kernel shapes according to spherical geometry (Tateno et al., 2018). Other methods, such as Flat2Sphere, learn spherical convolutions that mimic planar CNN responses on 360° imagery (Su & Grauman, 2017). SphereNet projects features between the plane and the sphere to maintain approximate rotational robustness (Coors et al., 2018). BiFuse and UniFuse fuse ERP and cube-map representations (Wang et al., 2020; Jiang et al., 2021). HRDFuse refines this concept by collaboratively learning holistic and regional depth distributions (Ai et al., 2023). More recently, transformer-based approaches, such

as PanoFormer and EGFormer, operate on ERP inputs using geometry-aware local attention (Shen et al., 2022; Yun et al., 2023). SFSS explores single-frame semantic segmentation from multi-modal spherical images (Guttikonda & Rambach, 2024). Elite360D further enhances efficiency through a semantic- and distance-aware bi-projection fusion scheme (Ai & Wang, 2024). While projection-based methods benefit from established 2D architectures, their treatment of spherical geometry is approximate, and achieving principled equivariance to arbitrary 3D rotations remains challenging.

## 2.2. Spherical CNNs

Spherical CNNs generalize convolutions to signals in $\mathbb{S}^2$ using spherical harmonics and group convolutions, producing representations that are exactly equivariant to SO(3) actions of 3D rotations (Cohen et al., 2018; Esteves et al., 2018). Early works construct spherical convolutions using spherical harmonics or spectral transforms. More recent approaches utilize spatial-domain discretizations and gauge-equivariant formulations to facilitate efficient learning on icosahedral and other spherical grids (Cohen & Welling, 2016; Worrall et al., 2017; Cohen et al., 2019). HexRUnet performs orientation-sensitive semantic segmentation in icosahedral spherical discretizations (Zhang et al., 2019). UGSCNN approximates the sphere with an icosahedral mesh and defines convolutions using parameterized differential operators on unstructured grids (Jiang et al., 2019). SpherePHD employs a spherical polyhedron representation with specialized convolution and pooling operators that reduce sampling distortion (Lee et al., 2019). RectConv adapts CNN kernels for fisheye cameras without retraining, providing an alternative method to handle distortion (Griffiths & Dansereau, 2024). DISCO proposes discrete-continuous spherical convolutions that are SO(3)-equivariant (Ocampo et al., 2023).

## 2.3. Spherical Transformers

Equivariant self-attention has been explored in general geometric settings, including LieTransformer (Hutchinson et al., 2021) and SE(3)-Transformer (Fuchs et al., 2020). More recently, transformers have been adapted to spherical domains, integrating attention mechanisms with spherical geometric priors. HEAL-SWIN adapts the Swin Transformer to the HEALPix sampling in the sphere (Carlsson et al., 2024). SpRePE introduces a spherical geometry-aware positional encoding via a global spherical positional embedding (Anonymous, 2025). Attention in the Sphere defines continuous attention kernels in $\mathbb{S}^2$ by adopting integration with respect to the invariant Haar measure, resulting in SO(3)-equivariant attention maps (Bonev et al., 2025). SphereUFormer, particularly relevant to our work, integrates novel spherical local self-attention into a U-shaped

transformer for 360° perception and demonstrates strong performance (Benny & Wolf, 2025).

Despite these advances, most existing spherical transformers are not fully SO(3)-equivariant. Instead, they typically preserve only a subgroup symmetry, which limits the robustness to camera tilt and object reorientation. For example, SphereUFormer is equivariant only to horizontal (yaw) rotations. From a computational perspective, many formulations of spherical local attention rely on *location-dependent* local positional embeddings (LPEs) or alternatively achieve SO(3) equivariance through computationally expensive *location-dependent* integration operations. In SphereUFormer, for instance, the LPE mechanism samples a learnable bias from a table indexed by query-dependent spherical coordinates. Such designs introduce additional memory and computational overhead, making them more challenging to implement in practical applications. Our work focuses on local attention mechanisms on $\mathbb{S}^2$ and aims to realize an efficient, LPE-free formulation that is fully SO(3)-equivariant.

## 3. Method

### 3.1. Preliminaries

**Spherical signal discretization.** For an omnidirectional 360° signal, we associate a feature vector $x(\theta, \phi) \in \mathbb{R}^C$ with each viewing direction $(\theta, \phi)$, where $\theta \in [0, \pi]$ denotes the altitude and $\phi \in [0, 2\pi)$ denotes the azimuth. Each direction corresponds to a unit vector (surface normal) $\mathbf{u} \in \mathbb{S}^2 \subset \mathbb{R}^3$ with $\|\mathbf{u}\|_2 = 1$. We discretize the sphere by sampling a finite set of directions. Several sampling strategies have been proposed (e.g., UV-sphere, cube-sphere, hexasphere, and icosphere). Following SphereUFormer, we adopt an icosphere representation that offers (i) near-uniform point distributions, (ii) high symmetry, and (iii) a natural multi-resolution hierarchy. We index the discretized location with $i$. In the sequel, we consider $\mathbf{x}_i \in \mathbb{R}^C$ to denote a $C$-dimensional token feature produced by an input embedding or an intermediate layer. A spherical signal is therefore represented as a set of sample locations $\{\mathbf{u}_i\}_{i=1}^N$ and corresponding features $\{\mathbf{x}_i\}_{i=1}^N$.

**Vector conventions.** Throughout, we represent per-token quantities as *column vectors*. For $\mathbf{a}, \mathbf{b} \in \mathbb{R}^c, \mathbf{c} \in \mathbb{R}^d$, the inner product is $\langle \mathbf{a}, \mathbf{b} \rangle := \mathbf{a}^\top \mathbf{b}$ (scalar), while $\mathbf{a}\mathbf{c}^\top \in \mathbb{R}^{c \times d}$ denotes the outer product. As elaborated in the subsequent sections, our positional modulation mechanism employs rank-one linear operators constructed via outer products.

**Problem formulation.** We consider a feature extractor $f$ that consumes features defined on a spherical grid and outputs updated features on the *same* grid. We say $f$ is SO(3)-*equivariant* if rotating the entire input sphere results

in an output that is rotated in the same manner. Formally, let $\pi_R(i)$ denote the induced permutation of sample indices $i$ resulting from the action of a global rotation $R \in \mathrm{SO}(3)$ on the discrete spherical signal (i.e., the induced permutation of sample indices after rotating and resampling on the same grid). Then $f$ is $\mathrm{SO}(3)$-equivariant if the following conditions hold:

$$f\big(\{[\mathbf{u}, \mathbf{x}]_{\pi_R(k)}\}_{k=1}^N\big)_i = f\big(\{[\mathbf{u}, \mathbf{x}]_k\}_{k=1}^N\big)_{\pi_R(i)} \forall i \subset 1, ...N.$$

This property is desirable for applications such as geospatial modeling, robotics with arbitrary sensor orientations, and scientific datasets defined on the sphere, where the notions of axis or gravity are irrelevant.

However, as discussed in Sections 1 and 2, many spherical transformers are not fully $\mathrm{SO}(3)$-equivariant. For example, SphereUFormer is equivariant only to horizontal (yaw) rotations. Moreover, to encode local geometry, existing spherical local attention typically relies on *location-dependent* local positional embeddings (LPEs), such as sampling a learnable bias from a table at query-dependent coordinates. The position-dependent sampling of bias weights incurs additional memory and computational overhead. Our goal is to design a local attention mechanism that is (i) equivariant to global $\mathrm{SO}(3)$ rotations on spherical sampling grids and (ii) free from location-dependent LPEs.

### 3.2. $\mathbb{S}oLA$

We propose $\mathbb{S}oLA$ (Figure 1), an *LPE-free* $\mathrm{SO}(3)$-equivariant local (linear) attention mechanism that operates directly on spherical signals. The central mechanism for achieving $\mathrm{SO}(3)$-equivariance without reliance on LPEs (or other computationally intensive, location-dependent integration schemes) involves positional modulation of queries and keys using the *global position* (normals) of each token, which induces the geodesic distance between associated spherical points *as a consequence of their dot product*. Importantly, this modulation is computed from a *single* token direction, while still inducing a pairwise relative distance via dot products, as we demonstrate below.

$\mathbb{S}oLA$. Each input feature $\mathbf{x}_i \in \mathbb{R}^C$ is projected to query, key, and value vectors $\mathbf{q}_i, \mathbf{k}_j \in \mathbb{R}^D$ and $\mathbf{v}_i \in \mathbb{R}^C$. For clarity, lets assume the single attention head. $\mathbb{S}oLA$ computes attention by aggregating values within each neighborhood via

$$\mathbf{y}_i = \sum_{j \in \mathcal{N}(i)} \sigma_j\Big(\langle \tilde{\mathbf{q}}_i, \tilde{\mathbf{k}}_i \rangle_F\Big) \mathbf{v}_j, \tag{1}$$

where $\mathcal{N}(i)$ denotes the geodesic neighborhoods defined by the $M$ nearest samples to token $i$ on the sphere, and $\tilde{\mathbf{q}}, \tilde{\mathbf{k}} \in \mathbb{R}^{D \times 4}$ are modulated queries and keys based on their

associated spherical coordinates as follows:

$$\tilde{\mathbf{q}}_i = \mathbf{q}_i[b, w\mathbf{u}_i^\top] \qquad \tilde{\mathbf{k}}_j = \mathbf{k}_j[1, \mathbf{u}_j^\top] \tag{2}$$

where $b, w$ is a scalar and $\mathbf{u} \in \mathbb{S}^2$ is the associated normal vector corresponding to the spherical coordinates. Equation (2) modulates each query and key by lifting their features using the outer product of four-dimensional global position vectors.

---

**Remark-1 Attention score of $\mathbb{S}oLA$**

*The pre-softmax similarity between tokens $i$ and $j$ induced by $\mathbb{S}oLA$ factors into a content term $\langle \mathbf{q}_i, \mathbf{k}_j \rangle$ and a geometric term $\cos \delta_{ij}$, where $\delta_{ij}$ is the great-circle distance on $\mathbb{S}^2$.*

---

*Proof (sketch).* Let the pre-softmax similarity be

$$s_{ij} := \langle \tilde{\mathbf{q}}_i, \tilde{\mathbf{k}}_j \rangle.$$

Substituting the embeddings from (2), we obtain

$$\begin{aligned} s_{ij} &= \langle \mathbf{q}_i[b, w\mathbf{u}_i^\top], \mathbf{k}_j[1, \mathbf{u}_j^\top] \rangle_F \\ &= b(\mathbf{q}_i^\top \mathbf{k}_j) + w(\mathbf{q}_i^\top \mathbf{k}_j)(\mathbf{u}_i^\top \mathbf{u}_j), \\ &= \langle \mathbf{q}_i, \mathbf{k}_j \rangle (b + w\cos\delta_{ij}). \end{aligned}$$

The score $s_{ij}$ depends on $\langle \mathbf{q}_i, \mathbf{k}_j \rangle$ and $\delta_{ij}$. $\qquad \square$

---

**Remark-2 Equivariance of $\mathbb{S}oLA$**

*Let $R \in \mathrm{SO}(3)$ be a global rotation and let $\pi_R(i)$ denote the induced permutation of sample indices $i$. Writing $\mathbf{y}_i := f\big(\{[\mathbf{u}, \mathbf{x}]_k\}_{k=1}^N\big)_i$, we have*

$$\mathbf{y}_i' := f\big(\{[\mathbf{u}, \mathbf{x}]_{\pi_R(k)}\}_{k=1}^N\big)_i = \mathbf{y}_{\pi_R(i)} \quad \text{for all } i$$

---

*Proof (sketch).* Under the permutation $\pi_R$, the projected features permute equivariantly: $\mathbf{q}_i \mapsto \mathbf{q}_{\pi_R(i)}$, $\mathbf{k}_j \mapsto \mathbf{k}_{\pi_R(j)}$, and $\mathbf{v}_j \mapsto \mathbf{v}_{\pi_R(j)}$. Consequently, the content term is preserved up to index permutation, $\langle \mathbf{q}_{\pi_R(i)}, \mathbf{k}_{\pi_R(j)} \rangle$.

The geometric term depends only on inner products of directions. Since $R$ is orthonormal, $(R\mathbf{u}_a)^\top(R\mathbf{u}_b) = \mathbf{u}_a^\top \mathbf{u}_b$, and the permutation $\pi_R$ induced by resampling preserves neighborhood relations. Hence, $s_{\pi_R(i)\pi_R(j)} = s_{ij}$ and $\mathcal{N}(\pi_R(i)) = \{\pi_R(j) \mid j \in \mathcal{N}(i)\}$. Therefore, the attention weights and value aggregation are equivariant, yielding $\mathbf{y}_i' = \mathbf{y}_{\pi_R(i)}$. $\qquad \square$

In summary, $\mathbb{S}oLA$ injects global spherical position information into attention via the unit directions $\mathbf{u}_i$. This avoids location-dependent LPE computation (e.g., via a lookup table) while preserving great-circle geometry, yielding an $\mathrm{SO}(3)$-equivariant local attention operator. Notice that this modulation is computed from a *single* token direction while still inducing a pairwise relative-distance term through dot products, as expected.

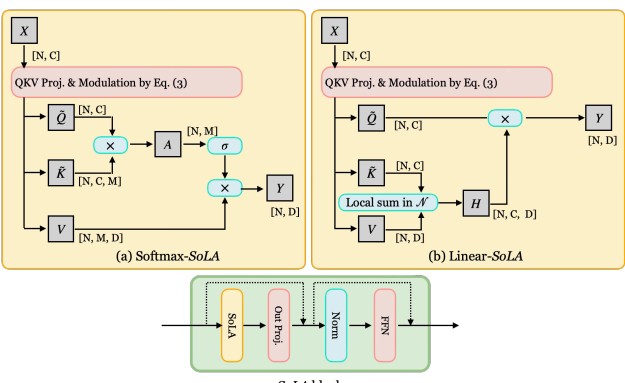

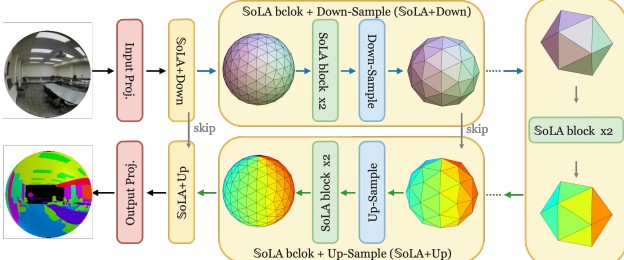

*Figure 3.* $\mathbb{S}oLA$ **U-Net.** A spherical signal is projected to feature, processed by a hierarchy of $\mathbb{S}oLA$ blocks (Figure 2) with down- and up- sampling and skip connections, and finally mapped to task outputs. This architecture follows SphereUFormer and replaces spherical local self-attention (SLSA) with $\mathbb{S}oLA$.

*Figure 2.* $\mathbb{S}oLA$ **block.** Input features are projected to $(Q, K, V)$, and modulated using direction vectors to obtain $\tilde{Q}, \tilde{K}$ using Equation (2) or (3). (a) Softmax-$\mathbb{S}oLA$ uses standard normalized attention within a local geodesic neighborhood. (b) Linear-$\mathbb{S}oLA$ first aggregates local key–value products to obtain $H$, then contracts with the query. Both variants are SO(3)-equivariant.

### 3.3. Efficient QK-first formulation

$\mathbb{S}oLA$'s embedding of Equation (2) quadruples the query/key dimension, thereby increasing the computation. The pre-softmax similarity score of $\mathbb{S}oLA$ can be reformulated as $\langle \mathbf{q}_i, \mathbf{k}_j \rangle \left( b + w \mathbf{u}_i^\top \mathbf{u}_j \right)$ (*Remark-1*). Therefore, instead of creating a modulated query/key with a fourfold larger dimension, one could adopt this formulation to compute query/key interactions. Specifically, it computes a similarity between the original query/key vector and the decay mask as $b + w \mathbf{u}_i^\top \mathbf{u}_j$ for $(i, j)$ similarity.

### 3.4. Input-dependent gating

The coefficient $b, w$ in the 4D modulation vector in (2) controls the weight of the geodesic distance term $\cos \delta_{ij}$. To increase $\mathbb{S}oLA$'s flexibility, we introduce input-dependent gating by adjusting the weights based on the inputs. Specifically, we modulate the query and key by incorporating *input-dependent channel-wise* gating as follows:

$$\tilde{\mathbf{q}}_i = \mathbf{q}_i \odot [\mathbf{b}_i^Q, \mathbf{w}_i^Q \mathbf{u}_i^\top], \quad \tilde{\mathbf{k}}_j = \mathbf{k}_j \odot [\mathbf{b}_j^K, \mathbf{w}_j^K \mathbf{u}_j^\top], \quad (3)$$

where $\mathbf{b}_i^Q, \mathbf{b}_j^K, \mathbf{w}_i^Q, \mathbf{w}_j^K \in \mathbb{R}^D$ represents the input dependent gating coefficient. The similarity $s_{ij}$ becomes:

$$s_{ij} = \langle \mathbf{q}_i \odot \mathbf{b}_i^Q, \mathbf{k}_j \odot \mathbf{b}_j^K \rangle + \langle \mathbf{q}_i \odot \mathbf{w}_i^Q, \mathbf{k}_j \odot \mathbf{w}_j^K \rangle (\mathbf{u}_i^\top \mathbf{u}_j). \quad (4)$$

This remains SO(3)-equivariant because the similarity depends only on inner products and the rotation-invariant geometric factor $\mathbf{u}_i^\top \mathbf{u}_j$. This variant incurs a minor additional computation to compute the channel-wise gating coefficient; however, it can be executed in parallel with the QKV-projection, resulting in negligible impact on latency. The increased representational power arises from the use of an asymmetric, token-dependent gating mechanism. We adopt this formulation in our experiments.

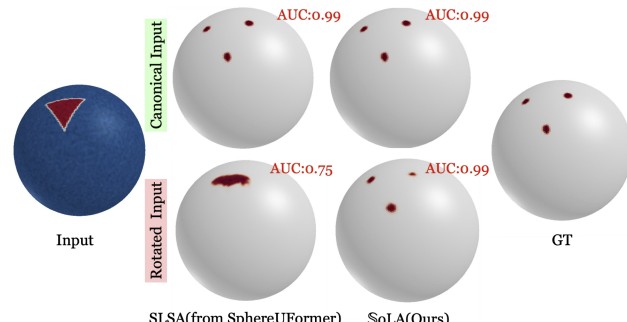

*Figure 4.* **SO(3)-equivariance test.** Corner detection. Top: Output from canonical input $f\left(\{[\mathbf{u}, \mathbf{x}]_k\}_{k=1}^N\right)_{\pi_R(i)}$. Both methods fit well with the training data. Bottom: Output from rotated input $f\left(\{[\mathbf{u}, \mathbf{x}]_{\pi_R(k)}\}_{k=1}^N\right)_i$. $\mathbb{S}oLA$ produces consistent outputs.

### 3.5. Linear $\mathbb{S}oLA$

By linearizing the softmax normalization, we can interchange the order of multiplication and first accumulate the products of the key-value pairs. Given the same position-embedded queries and keys $\tilde{\mathbf{q}}_i, \tilde{\mathbf{k}}_j$, we define *linear-$\mathbb{S}oLA$*:

$$\mathbf{y}_i = \sum_{j \in \mathcal{N}(i)} (\tilde{\mathbf{q}}_i^\top \tilde{\mathbf{k}}_j) \mathbf{v}_j = \left( \sum_{j \in \mathcal{N}(i)} \mathbf{v}_j \tilde{\mathbf{k}}_j^\top \right) \tilde{\mathbf{q}}_i. \quad (5)$$

The positional modulation depends solely on each token's own direction $\mathbf{u}_j$, which does not rely on query-specific relative offsets or on other tokens and their positions. Consequently, we can compute the keyvalue accumulation through efficient local summation without explicitly materializing per-query neighbors. This approach significantly reduces memory usage, particularly when modeling relationships involving a large number of local neighborhoods, including the full-attention case (Section 4.4).

### 3.6. $\mathbb{S}oLA$ U-Net

The softmax and linear variants of the $\mathbb{S}oLA$ block adhere to the standard transformer-block design, as illustrated in Figure 2. We integrate $\mathbb{S}oLA$ into the U-shaped encoderdecoder architecture of SphereUFormer by replacing its *Spherical*

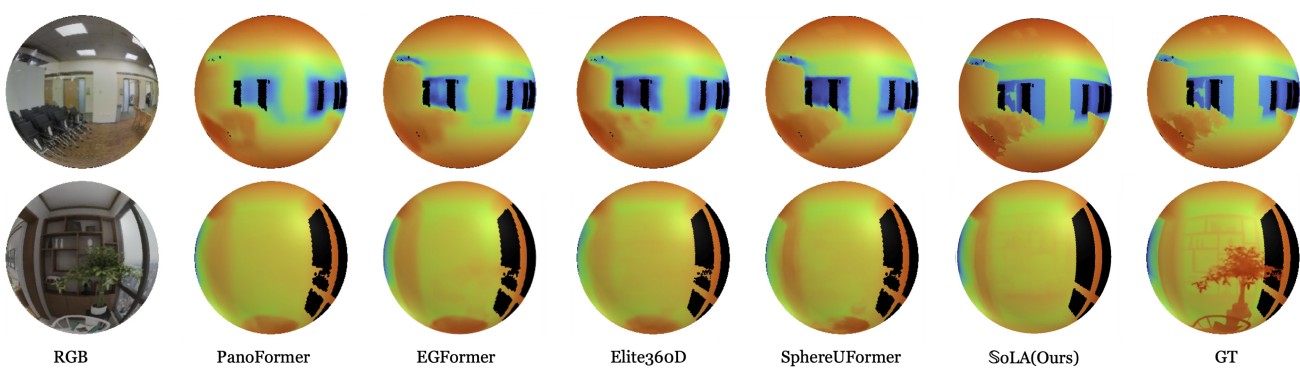

Figure 5. **Depth Estimation.** Top: Stanford2D3D. Bottom: Structured3D.

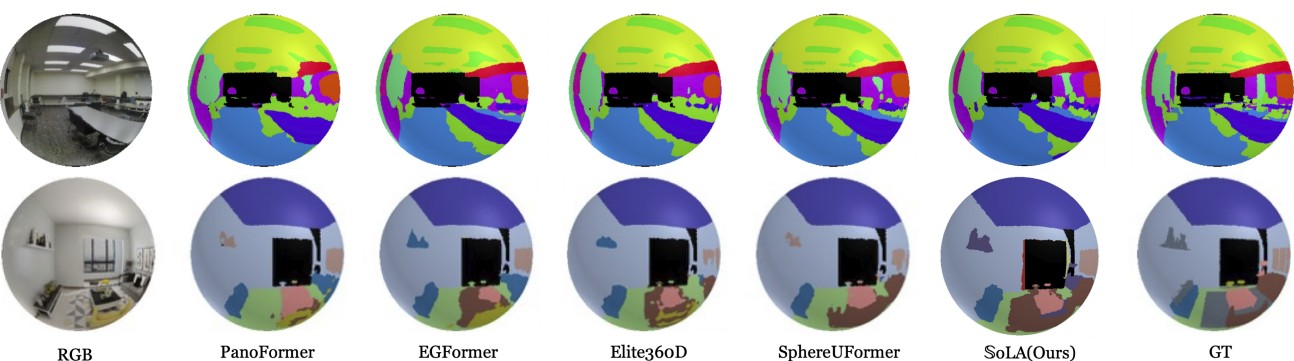

Figure 6. **Semantic Segmentation.** Top: Stanford2D3D. Bottom: Structured3D.

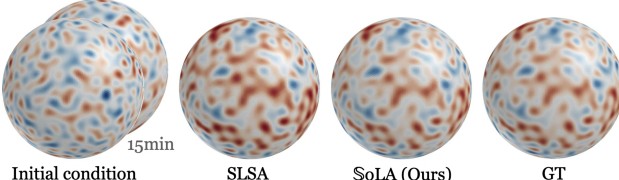

Figure 7. **PDE.** Shallow water equations (lead time of 1 hour).

*Local Self Attention* (SLSA) with $\mathbb{S}oLA$, as illustrated in Figure 3. Each encoder stage applies two $\mathbb{S}oLA$ transformer blocks, followed by a down-sampling operator that coarsens the spherical grid. The decoder mirrors the encoder, utilizing up-sampling operators and skip connections that concatenate encoder features at matching resolutions.

## 4. Experiments

We first verify the SO(3)-equivariance of $\mathbb{S}oLA$ on a controlled toy corner-detection task (Section 4.1). We then evaluate performance on practical applications: depth estimation (360°), semantic segmentation (Section 4.2), forecasting water dynamics on the sphere (Section 4.3), and spherical image classification (Section 4.4). Finally, we present ablations to quantify the impact of important design choices (Section 4.6). Our experimental setup mostly follows that of the SphereUFormer (Benny & Wolf, 2025). Additional implementation details are provided in the supplementary material.

### 4.1. SO(3)-equivariance test

This experiment numerically evaluates the SO(3)-equivariance of $\mathbb{S}oLA$ in a controlled corner-detection task. The training set consists of a *single* spherical image of a triangle, for which we supervise the corner responses. At test time, we evaluate accuracy by randomly rotating the sphere. Our network comprises three $\mathbb{S}oLA$ blocks (Figure 2) and is compared against a baseline using SLSA (local attention from SphereUFormer). Figure 4 demonstrates that $\mathbb{S}oLA$ produces outputs that rotate consistently with the input, as expected from its SO(3) equivariant formulation.

### 4.2. Omnidirectional scene understanding

This experiment evaluates 360° depth estimation and semantic segmentation on Stanford2D3D (Armeni et al., 2017) and Structured3D (Zheng et al., 2019). To isolate the effect of the proposed attention mechanism, we adopt $\mathbb{S}oLA$ U-Net (Section 3.6), which preserves the SphereUFormer architecture (down/up-sampling, skip connections, normalization, and MLP blocks, etc.) while replacing *only* the spherical attention module SLSA with $\mathbb{S}oLA$. We adhere to the SphereUFormer training protocol, training for 400 epochs on Stanford2D3D and 120 epochs on Structured3D using the Adam optimizer (Kingma & Ba, 2015) without any scheduling. To assess robustness to arbitrary 3D rotations, we also evaluate rotated variants of each dataset

*Table 1.* **Omnidirectional perception.** For SphereUFormer and $\mathbb{S}oLA$ U-Net, the results from rotated input, Rotated-{Stanford2D3D, Structured3D}, are shown after the canonical results. Input rank 7 (256x512 for ERP method). Time ([ms]) evaluated on Nvidia H200.

| Model | FLOPS | Time | Depth Estimation | | | | | | Semantic Segmentation | | | |
| | | | Stanford2D3D | | | Structured3D | | | Stanford2D3D | | Structured3D | |
| | | | MAE↓ | MRE↓ | δ₁↑ | MAE↓ | MRE↓ | δ₁↑ | Acc.↑ | mIoU↑ | Acc.↑ | mIoU↑ |
|---|---|---|---|---|---|---|---|---|---|---|---|---|
| PanoFormer | 11.8G | - | .174 | .078 | 92.5 | .154 | .051 | 94.8 | 83.1 | 60.6 | 94.9 | 49.7 |
| EGForme | 15.6G | - | .170 | .075 | 93.1 | .150 | .049 | 95.2 | 86.5 | 66.4 | 95.0 | 51.5 |
| SFSS | 18.9G | - | .179 | .081 | 92.2 | .155 | .051 | 95.0 | 86.9 | 68.2 | 95.2 | 51.9 |
| HexRUnet | 12.4G | - | .201 | .090 | 90.1 | – | – | – | 81.7 | 56.1 | – | – |
| HEAL-SWIN | 39.0G | - | .189 | .084 | 92.2 | – | – | – | 85.5 | 63.2 | – | – |
| Elite360D | 13.6G | - | .169 | .069 | 93.5 | .147 | .046 | 95.9 | 87.4 | 71.4 | 95.3 | 52.0 |
| SphereUFormer | 14.9G | 95.0 | .165 1.05 | .071 0.48 | 94.0 28.9 | .147 .740 | .045 .530 | 96.4 25.5 | 88.6 37.4 | 72.2 15.3 | 95.8 34.0 | 53.0 6.94 |
| $\mathbb{S}oLA$ | 19.4G | 94.2 | .166 .169 | .071 .070 | 94.0 93.3 | .148 .149 | .047 .447 | 95.9 96.1 | 88.4 88.4 | 72.2 72.2 | 95.4 95.2 | 53.0 53.1 |

*Table 2.* **PDE.** Shallow water equations forecasting. Input rank 6.

| Method | Train | | | Test | | |
| | Loss ↓ | L1 ↓ | L2 ↓ | Loss ↓ | L1 ↓ | L2 ↓ |
|---|---|---|---|---|---|---|
| SLSA | 0.0795 | 0.0109 | 0.0328 | 0.1322 | 0.0324 | 0.0972 |
| $\mathbb{S}oLA$ | 0.0873 | 0.0131 | 0.0392 | 0.0870 | 0.0130 | 0.0389 |

*Table 3.* **Spherical image classification.** Trained on the canonical training set, and report the test accuracy for the randomly rotated test set. Input rank 4. Time ([ms]) evaluated on Nvidia H200.

| Method | Accuracy [%] | | | Compute | |
| | Train | Canonical | Rotated | FLOPS | Time |
|---|---|---|---|---|---|
| Linear GPA | 99.4 | 96.7 | 15.3 | 5.9G | 2.05 |
| Linear-$\mathbb{S}oLA$ | 99.1 | 96.4 | 94.3 | 4.1G | 1.66 |

*Table 4.* **Ablations on positional encoding methods.** The semantic segmentation on (rotated) Stanford2D3D, and classification on Shperical-MNIST. +Vertical PE uses vertical PE from SphereUFormer. −ID gate uses learnable gates instead of the input-dependent gates (Section 3.4). −$\mathbb{S}oLA$+SLSA is SphereUFormer in Table 1. −$\mathbb{S}oLA$+GPE is Linear GPA in Table 3.

| Variant | Segmentation mIoU ↑ | | Classification Acc. ↑ | |
| | Canonical | Rotated | Canonical | Rotated |
|---|---|---|---|---|
| Full $\mathbb{S}oLA$ | 71.9 | 71.0 | 96.4 | 94.3 |
| +Vertical PE | 73.4 | 52.6 | 96.9 | 72.3 |
| −ID gate | 70.8 | 69.9 | 95.3 | 93.9 |
| −$\mathbb{S}oLA$ | 53.2 | 52.9 | 36.4 | 33.5 |
| −$\mathbb{S}oLA$+SLSA/GPE | 72.2 | 15.3 | 96.7 | 15.3 |

by applying a random global $SO(3)$-rotation to each test panorama without training on the rotated data. We report the Mean Absolute Error (MAE), Mean Relative Error (MRE), and $\delta_1$ accuracy for depth, as well as pixel accuracy and the mean Intersection over Union (mIoU) for segmentation; these are the default metrics for each evaluation.

Table 1 summarizes quantitative comparisons, while Figure 5-6 provides qualitative results. Figure 8 visualizes representative outputs under rotation. Across both tasks, $\mathbb{S}oLA$ U-Net achieves comparable accuracy to SphereUFormer on canonical inputs. For the zero-shot novel rotated input, $\mathbb{S}oLA$ produces consistent predictions under global rotations, reflecting the intended $SO(3)$-equivariant design, whereas the non-equivariant methods' accuracy deteriorates significantly. Our method is comparable to the baseline but shows a slight lag. We hypothesize that this is mainly attributed to the nature of the task, which benefits from the gravity cue. As we demonstrate in the ablation study, $\mathbb{S}oLA$ with vertical PE performs slightly better than the baseline, *although it breaks the* $SO(3)$ *symmetry* (Table 4).

**Baselines.** Since our model employs the same U-shaped network as SphereUFormer and differs only in its attention

mechanism, it serves as the most direct baseline. We additionally compare against representative projection-based and spherical-grid methods. Projection-based: PanoFormer (Shen et al., 2022), EGFormer (Yun et al., 2023), SFSS (Guttikonda & Rambach, 2024), and Elite360D (Ai & Wang, 2024). Spherical-grid: HexRUnet (Zhang et al., 2019) and HEAL-SWIN (Carlsson et al., 2024).

### 4.3. Forecasting partial differential equations on sphere

This task forecasts the dynamical evolution governed by the shallow water equations on a spinning sphere, a widely adopted benchmark for data-driven learning of partial differential equation dynamics on the sphere (Bonev et al., 2025; Liu-Schiaffini et al., 2024). To probe full-$SO(3)$-equivariance, we evaluate forecasts under random global $SO(3)$-rotations of the axes of the spin. To allow the model to estimate the spin axis implicitly, we provide the states at 15-minute intervals from the start, along with the initial states. We utilize a four-stack of $\mathbb{S}oLA$ blocks without any down- or up-sampling and compare it against methods that adopt SLSA under the same architecture and training protocol. Table 2 reports quantitative results, while Figure 7 visualizes representative rollouts. Since the underlying dynamics are rotation-equivariant, this task directly benefits from $\mathbb{S}oLA$'s $SO(3)$-equivariance, as demonstrated.

### 4.4. Spherical digit classification by global attention

The task is to predict the digit label from a spherical image. We used Spherical MNIST (Esteves et al., 2020) for this task. To assess rotation robustness, we trained on the canonical training set without rotation augmentation and reported classification accuracy under random test-time SO(3)-rotations. To highlight the efficiency of the linear formulation of (5), we evaluate it in the full linear-attention setting, where the neighborhood encompasses the entire sphere; i.e., $\mathcal{N}(i)$ in (5) contains all tokens. In this regime, softmax attention, which has quadratic complexity with respect to the number of tokens, incurs substantial memory usage and FLOPS, even at a coarse discretization of 2,562 tokens for rank 4, whereas linear attention scales linearly. Our model consists of four linear $\mathbb{S}oLA$ blocks, followed by a classifier that employs global average pooling and a single MLP head. Since SLSA is incompatible with the linear formulation, we compare it to a baseline that utilizes a global positional embedding (GPE) for token positions. Table 3 reports accuracy on both canonical and randomly rotated test sets. Since $\mathbb{S}oLA$ is SO(3)-equivariant by design, it achieves zero-shot generalization to unseen digit orientations.

### 4.5. Computation

We compare FLOPS and latency in both the softmax- (Table 1) and linear-attention (Table 3) settings. In the softmax-attention setting, the $\mathbb{S}oLA$-based network exhibits higher FLOPS than the SLSA-based baseline (SphereUFormer) at the same feature embedding dimension $D$. Nevertheless, it achieves slightly lower latency in practice because it avoids the costly, albeit low FLOPS, location-dependent LPE operation. In the linear setting, $\mathbb{S}oLA$ maintains comparable FLOPS and latency while providing substantially stronger zero-shot rotation robustness compared to a baseline.

### 4.6. Ablations

**Global positional cue.** $\mathbb{S}oLA$ is SO(3)-equivariant, which is crucial for strong robustness against global rotations and facilitates zero-shot generalization across camera and object rotations, as demonstrated in Table 4. However, some 360° perception tasks benefit from a privileged notion of "up" (e.g., gravity). To study this trade-off, we inject a global vertical positional embedding (the same cue used in SLSA) into the input features, intentionally disrupting SO(3)-equivariance. As shown in Table 4 (+ vertical PE), this slightly enhances performance on canonical orientations but degrades robustness under test-time rotations, as expected.

**Distance-preserving positional embedding.** We evaluate the contribution of the proposed distance-preserving positional embedding by removing it from $\mathbb{S}oLA$. As shown in Table 4 (- $\mathbb{S}oLA$), this change results in a substantial drop in

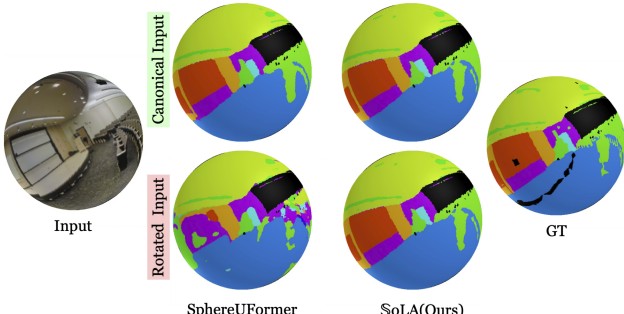

*Figure 8.* **Zero-shot estimation on rotated input.** Top: Rotated results of canonical input $f\big(\{[\mathbf{u}, \mathbf{x}]_k\}_{k=1}^N\big)_{\pi_R(i)}$ (same as training). Bottom: Results from rotated input $f\big(\{[\mathbf{u}, \mathbf{x}]_{\pi_R(k)}\}_{k=1}^N\big)_i$. $\mathbb{S}oLA$ produces outputs that rotate consistently with the input although it is not training on rotated data. Stanford2D3D segmentation task.

accuracy. Without this embedding, local attention becomes permutation-invariant within each neighborhood; therefore, it cannot capture structured local patterns on the sphere.

## 5. Summary, Limitations, and Future Work

We introduce $\mathbb{S}oLA$, an SO(3)-equivariant LPE-free local attention mechanism for spherical signals. By modulating the query and key features using their associated positions on the sphere, the dot product of the modulated query and key induces content similarity and their geodetic distance. Experiments demonstrate zero-shot generalization to novel rotations while remaining computationally efficient. Below, we discuss the current limitations and future directions.

**Gravity cue.** $\mathbb{S}oLA$ is equivariant to arbitrary global SO(3) rotations, which is the main focus of this study, and it is crucial for the demonstrated rotational robustness. However, there are many real-world settings where the direction of gravity is informative (e.g., indoor panoramas are typically gravity-aligned), and a model may benefit from using absolute elevation or orientation cues. As shown in the ablation (Section 4.6), injecting a global positional cue can improve performance on canonical orientations while degrading robustness under test-time rotations. A promising direction would be to inject the gravity cue (Refer to supplement-S2 for more discussion on this point.).

**Operating on ERP or fisheye formats.** In this work, we evaluate $\mathbb{S}oLA$ on signals resampled to an icosphere grid. In practice, 360° imagery is often stored as ERP or fisheye camera frames. We consider the Linear-$\mathbb{S}oLA$ formulation of Equation (5) to be best suited for this case; with the associated normal directions for each pixel, one can compute the linear attention by deforming the average kernel. One attractive direction is to explore this approach, exploiting $\mathbb{S}oLA$ for 2D data without resampling to a mesh.

## Impact Statement

This work introduces SoLA, an SO(3)-equivariant local attention mechanism for spherical signals. The primary contribution is methodological: improving the robustness and efficiency of neural networks operating on omnidirectional data. We expect this work to benefit applications involving spherical or wide-field observations, including robotics, autonomous driving, Earth observation, meteorology, scientific computing, and medical imaging. By improving robustness to viewpoint and orientation changes, the proposed method may reduce the amount of task-specific engineering and data augmentation required to deploy machine learning systems in such settings.

An additional potential benefit is the ability to more naturally integrate data originating from different camera models. Spherical representations provide a common geometric domain for perspective, fisheye, and omnidirectional cameras. Consequently, methods such as SoLA may facilitate future vision foundation models that can be trained on data collected from heterogeneous camera systems while sharing a unified representation. Such capability could improve data efficiency and broaden the applicability of learned visual representations across domains and sensor configurations.

The proposed method does not introduce application-specific risks beyond those generally associated with machine learning systems. As with other perception models, downstream deployments in safety-critical domains such as autonomous vehicles or robotics require rigorous validation and testing. Furthermore, the ability to learn from large-scale visual data can raise concerns regarding privacy, data governance, and responsible data collection practices. We believe that these considerations should be addressed at the data set and deployment levels. Overall, we expect the social impact of this work to be positive, primarily by enabling more reliable and flexible machine learning models for spherical and omnidirectional data.

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

# Supplemental Material for
## *Spherical SO(3) Equivariant Local Attention*

## S1. Experimental Details

In this Supplementary Section, we provide details of the experimental setup and implementation that do not fit within the main paper. Our codebase is based on the official repository of SphereUFormer(Benny & Wolf, 2025) (`https://github.com/yanivbenny/sphere_uformer`). We adopted the same data processing, data augmentation, training schedules, and evaluation protocols as in the original implementation, unless otherwise stated.

### $360°$ Scene Understanding

This experiment evaluates $360°$ monocular depth estimation and semantic segmentation from equirectangular panoramas, including robustness to unseen 3D rotations during testing.

#### DATASET

We utilize Stanford2D3D (Armeni et al., 2017) and Structured3D (Zheng et al., 2019). Structured3D provides three scene configurations (*full*, *simple*, and *empty*) based on the number of rendered furniture items; we utilize the *simple* configuration. As noted in the official repository (`https://github.com/bertjiazheng/Structured3D`), some scenes are invalid, and we also encountered a small number of corrupted files. We excluded these files from all experiments. We utilized the official sprite training and testing datasets for both Stanford2D3D and Structured3D.

To evaluate robustness to 3D rotations, we construct rotated test sets for both datasets. Given an ERP panorama, we sample a random rotation matrix $\mathbf{R} \in SO(3)$ and apply it to the associated surface normals on the ERP grid. We then resample the panorama using the rotated sampling grid. We refer to these test sets as Rotated-Stanford2D3D and Rotated-Structured3D.

#### NETWORK

For SphereUFormer, we employ the original architecture without modification: we utilize four down-sampling stages, a bottleneck, and four up-sampling stages, each incorporating two spherical transformer blocks that implement Spherical Local Self-Attention (SLSA).

To isolate the effect of the proposed attention mechanism, we utilize the $\mathbb{S}oLA$ U-Net (Section 3.6), which preserves the SphereUFormer architecture, including down-sampling, up-sampling, skip connections, normalization, MLP blocks, etc., while replacing only the spherical attention module (SLSA) with $\mathbb{S}oLA$.

The input to the network is a feature discretized by an icosphere of rank 7, sampled from the $256 \times 512$ panorama (ERP) image. Because the network includes initial down-sampling and final up-sampling layers, the transformer blocks operate on ranks $\{6, 5, 4, 3\}$.

Other settings also follow the SphereUFormer. The number of neighborhoods $M$ is 19 (i.e, "window coefficient" $C_{win} = 2$) across all layers. The hidden dimension for the first layer is set to 32 and increased to $32 \times 2^l$ for the encoder layer index $l$. Other parameters remain consistent with their official configuration.

#### TRAINING

We also adhere to the SphereUFormer training protocol: 400 epochs on Stanford2D3D and 139 epochs on Structured3D. We optimize using the Adam optimizer (Kingma & Ba, 2015) with a learning rate of $10^{-4}$ (without weight decay) and a batch size of 16. We utilize BerhuLoss (Laina et al., 2016) for depth estimation and Cross-Entropy loss for semantic segmentation.

#### EVALUATION

*Depth estimation.* We report the Mean Absolute Error (MAE), Mean Relative Error (MRE), and $\delta_1$ accuracy. For Stanford2D3D, SphereUFormer is reported to evaluate depth up to $10\,\text{m}$ as detailed in the paper; however, following the official implementation (`https://github.com/yanivbenny/sphere_uformer/`), we cap the depth at $5.12\,\text{m}$

and reproduce comparable accuracy under this setting. Evaluation on the rotated test sets follows the same procedure as that on the canonical test sets.

*Semantic segmentation.* We report pixel accuracy and the mean Intersection-over-Union (mIoU) for semantic segmentation. The regions where the depth exceeds the specified threshold are excluded.

We report the runtime for a single batch evaluated on the NVIDIA H200 GPU.

## Forecasting partial differential equations on sphere

This experiment evaluates the forecasting of spherical dynamics for partial differential equations (PDEs). We use the shallow-water equations (SWE) on a rotating sphere for this purpose. It models a thin layer of fluid covering a rotating sphere $\mathbb{S}^2$.

### DATASET

Shallow water equations (SWE) on $\mathbb{S}^2$ serve as a standard benchmark for learning PDE dynamics on the sphere (Bonev et al., 2025; Liu-Schiaffini et al., 2024). To create ground truth data, we use the open-source SWE simulator and solver (https://github.com/NVIDIA/torch-harmonics/tree/main/examples/shallow_water_equations). The training data for the SWE is generated by randomly sampling initial conditions and advancing them over time using a classical numerical solver. The initial geo-potential height and velocity fields are modeled as Gaussian random fields on the sphere. The parameters of the PDE, including gravity, the sphere's radius, and angular velocity, are selected to reflect Earth's conditions.

To probe full-SO(3) equivariance, we have developed a new protocol. The solution (i.e., the state 1 hour ahead) depends on the rotation axis along which the water is positioned. In the original protocol (Bonev et al., 2025; Liu-Schiaffini et al., 2024), the axis is fixed; therefore, full-SO(3) equivariance is not required. To evaluate robustness to full-SO(3) rotation, we randomly rotated the axis. Without knowledge of the axis, estimating the future state is infeasible. To enable the model to estimate the axis implicitly, we concatenate states at 15-minute intervals as additional input.

### NETWORK

We utilize a 4-block structure of $\mathbb{S}oLA$ blocks ((Figure 2)) without down-sampling or up-sampling, and we compare this approach against methods that adopt SLSA, the local attention mechanism from SphereUFormer, while keeping the other modules unchanged.

### TRAINING

Training data are generated on the fly using a spectral method to numerically solve the PDE on an equiangular grid with a spatial resolution of 256 512 and time steps of 150 seconds up to 1 hour ahead, serving as the ground truth target for the spherical network estimation. We trained 51,200 randomly generated samples. During training, we turned off random axis rotation. We used the Adam optimizer with an initial learning rate of $10^{-4}$ and applied cosine decay. We employed squared L2 loss, which is the default choice for this task. Both networks are trained with exactly the same protocol.

### EVALUATION

During testing, we randomly rotated the spin axis and evaluated it using 512 novel samples. We report the $L_1$ and $L_2$ norms, along with the squared $L_2$ losses employed during training.

The visual results presented in Figure 7 illustrate a one-hour forecast generated from a single forward pass of the model. Longer forecast horizons can be obtained by employing an autoregressive procedure, in which the model's own predictions are iteratively fed back as inputs for subsequent forecasting steps.

## Spherical Digit Classification with Full Attention

The task is to predict the digit label from a spherical image. To assess rotation robustness, we train on the canonical training set without rotation augmentation and report classification accuracy under random test-time SO(3)-rotations.

DATASET

We utilized Spherical MNIST (Esteves et al., 2020) for this task. To create the spherical MNIST data, we employed the codebase from s2cnn (https://github.com/jonkhler/s2cnn). We modified their code to output the equirectangular image, which is compatible with the SphereUFormer data loader. The number of generated training and test samples matches that of the original MNIST dataset. We removed "9" to eliminate ambiguity with "6" when rotated by 180 . Therefore, the actual numbers of training and testing samples are approximately 10% lower than those of the original dataset.

NETWORK

To highlight the efficiency of the linear formulation, we evaluate it in the *full* linear-attention setting, where the neighborhood covers the entire sphere; i.e., $\mathcal{N}(i)$ in Eq. (5) includes all tokens. In this regime, softmax attention incurs substantial memory usage due to its quadratic complexity with respect to the number of input tokens, even at a coarse discretization (rank 4; 2,562 points = 2,562 tokens), whereas linear attention scales linearly with the number of input tokens.

Our model consists of four linear $\mathbb{S}oLA$ blocks (Figure 2), followed by a classifier that employs global average pooling and a single MLP head. Since SLSA (the local attention mechanism from SphereUFormer) is not compatible with the linear formulation, we compare it with a baseline that uses a global positional embedding (GPE) with latitude and longitude as embedding features. We adopt the standard GPE method, encoding the global location $\mathbf{u} \in \mathbb{R}^3$ (corresponding to latitude and longitude) using a learnable Fourier positional embedding layer, which comprises a single MLP with the same dimensionality as the hidden layer. This embedding is added to the incoming features before the QKV projection.

In this experiment, we adopt different embedding dimensions for our method and the baseline to approximate their computational complexities (FLOPS and latency, as shown in Table 3). Our embedding dimension is set to 128, whereas the baseline employs 256.

TRAINING

During training, we turned off random axis rotation. We train for 50 epochs using the Adam optimizer with an initial learning rate of $10^{-4}$ and applied cosine decay. We utilize the standard cross-entropy loss for training. Both networks are trained using the same protocol. To assess zero-shot rotation robustness, we train on the canonical training set without rotation augmentation and report classification accuracy under random test-time SO(3)-rotations.

EVALUATION

We report classification accuracy under random test-time SO(3)-rotations as well as training accuracy. Note that "9" is not included in the test data. We also report the runtime for a single batch evaluated on the NVIDIA H200 GPU.

**Remarks on the Architecture and Hyperparameter Optimization**

Our network design adheres to SphereUFormer, isolating the effects of the attention mechanism. Consequently, the overall architecture depth, width, and neighborhood size may be suboptimal for certain tasks. Exploring task-specific architectural choices and hyperparameter tuning—potentially guided by efficiency constraints—represents an important direction for future work.

## S2. Injecting the gravity cue

One way to improve accuracy is to use the minimum equivariance for the target applications. The full SO(3) equivariance might be too strict for some applications. For example, a chair is usually upright, and using this prior can help the model recognize the object if gravity is known. In this case, restricted SO(3)-equivariance–where the model remains equivariant under longitudinal/latitudinal movement but sensitive to local in-plane rotation–would be a good choice. An extension of SoLA that induces an additional angle-to-gravity term, $\beta$, would achieve this equivariance by the following modulation:

$$[b, wu^\top + w_g u_g^\top + w_h u_h^\top],$$

where $u_g$ and $u_h$ are the local gravity and horizontal vector.

When $w_g$ and $w_h$ are zero, this reduces to the current formulation (Equation (2)), inducing the geodetic distance $\delta$, which is SO(3)-equivariant. When they are nonzero, the query key dot product induces a $\beta$ term of the form.

$$A \cos \delta + B \sin \delta \cos \beta + C \sin \delta \sin \beta.$$

This extension allows the model to exploit the gravity cue while retaining equivariance to longitudinal/latitudinal movement (restricted SO(3)-equivariance). By making $w_g$ and $w_h$ input-dependent, the model can adaptively fuse gravity cues when they are helpful, while still being able to fall back on strict SO(3)-equivariance when they are not.

**Note 1.** SphereUFormer and the SLSA variants in the ablation (Table 4) lack this property. Those methods directly embed the global vertical position; they can use the gravity cue, but their functions are also sensitive to latitudinal movement.

**Note 2.** Many applications do not have a preferred axis, such as our SWE forecasting and medical imaging. Moreover, even for gravity-aware applications, strict SO(3)-equivariance would be more efficient at capturing low-level visual cues, as these cues may appear in arbitrary orientations. Therefore, a hybrid architecture that adopts full SO(3) equivariance in early layers and SE(2) equivariance in later layers would be a good design choice for specific applications. The current study fucus on the strict SO(3)-equivariance; therefore, we left the exploration in this direction for future work.

## S3. Computational complexity comparison for softmax and linear variant of $\mathbb{S}oLA$

We chose Softmax-$\mathbb{S}oLA$ for the depth and SS tasks because it is more efficient, in both FLOPs and activation memory, than the linear variant when $M = 19$.

In the KV-first formulation, Linear-$\mathbb{S}oLA$ avoids explicitly materializing neighbor key/value pairs, so its neighbor-memory cost is constant with respect to $M$. However, in the KV-first case, it cannot use the efficient technique that bypasses 4D lifting (Section 3.3), which applies only to the QK-first formulation. So, there is a tradeoff between the two approaches: QK-first avoids 4D lifting but requires neighbor materialization, whereas KV-first does the opposite.

Following the SphereUFormer's settings, we set $C_{win} = 2$ ($M = 19$). In this setting, the QK-first formulation, which avoids 4D lifting, is more efficient; the cost of 4D lifting hinders the advantage of avoiding neighbor materialization at this neighborhood size.

The break-even point lies between $C_{win} = 2$ ($M = 19$) and $C_{win} = 3$ ($M = 43$). For $C_{win} \geq 3$, Linear-$\mathbb{S}oLA$ requires less activation memory because the gain from avoiding explicit materialization of key/value pairs outweighs the increase from the 4D lifting. However, at $M = 43$, the accuracy of both SphereUFormer and Softmax-$\mathbb{S}oLA$ dropped in the tasks. For example, on the Stanford2D3D SS task, mIoU is 71.5 and 71.6 for SphereUFormer and Linear-$\mathbb{S}oLA$, respectively. We therefore used $M = 19$, which matches the default of SphereUFormer. In higher-resolution settings, $C_{win} \geq 3$ might yield better accuracy, but we have not tested this setting.

At $C_{win} = 2$, the accuracy of Softmax-$\mathbb{S}oLA$ and Linear-$\mathbb{S}oLA$ is very similar. In the Stanford2D3D SS task, the mIoU of Linear-$\mathbb{S}oLA$ is 72.1 (Softmax-$\mathbb{S}oLA$ is 72.2). We chose QK-first for efficiency and adopted softmax to compare with SphereUFormer.

The extreme case is $M = N$, which corresponds to full attention. In this setting, the difference is significant: the activation memory of the QK-first formulation (Softmax-$\mathbb{S}oLA$, or SphereUFormer) increases with $\mathcal{O}(N^2)$, easily exceeding the GPU's memory limits, while in the KV-first formulation, it is $\mathcal{O}(N)$. The image classification experiments in Section 4.4 demonstrate the effectiveness of Linear-$\mathbb{S}oLA$ in this extreme setting. In Table 3, we report results using a low-resolution input of icosphere rank 4 (2.5K points), roughly reflecting the resolution of the original MNIST data (28x28). We can increase this to rank 7 (328K points, same settings as the depths and SS task in Table 1) or even higher. We conducted the experiments with input rank 7 using Linear-$\mathbb{S}oLA$. The accuracies were 96.5% and 94.2% for the canonical and rotated inputs (no significant improvement).

## S4. Detailed discussion on computational complexity and memory footprint

The discussion of memory requirements and computational complexity differs between Softmax-$\mathbb{S}oLA$ and Linear-$\mathbb{S}oLA$.

**Softmax-$\mathbb{S}oLA$** In the Softmax-$\mathbb{S}oLA$ setting used for the dense prediction experiments (depth and semantic segmentation), SphereUFormer consumes less activation memory in our settings (i.e., the same QKV feature dimension), mainly because

SoLA adds input-dependent gating to compute the decay mask $M$ (Section 3.4). For this, we first compute the query-key dot product of the unmodulated vectors, then construct the decay mask $M \in \mathbb{R}^{N \times M}$ using the input-dependent gating and normals as:

$$M_{ij} = [b_i^Q, w_i^Q \mathbf{u}_i^\top] \times [b_j^K, w_j^K \mathbf{u}_j^\top]^\top,$$

and apply that decay mask to the attention logits. The $M$ nearest neighbors of $b^K$ and $w^K$ are computed in the same way as the neighbor keys and values using the pre-computed indices. These would be a memory and computational overhead compared with the prior method.

The memory advantage of Softmax-$\mathbb{S}oLA$ over the prior method is that it removes the position-dependent sparse bilinear sampling used to build local positional embedding biases, thereby avoiding irregular memory access and improving hardware friendliness. In terms of the computation, LPE-based methods (e.g., SphereUFormer) compute attention bias using local neighbors, which requires sparse, position-dependent, bilinear sampling. Softmax-$\mathbb{S}oLA$ avoids this neighbor operation by the novel position modulation.

**Linear-$\mathbb{S}oLA$**   The Linear-$\mathbb{S}oLA$ inherits both the memory and computational efficiency discussed above. In addition to the above benefit, Linear-$\mathbb{S}oLA$ (KV-first) avoids explicit per-query neighbor materialization, so the memory required for neighbor key/value features no longer grows with $M$. In this case, the activation memory of SoLA would be smaller than that of SphereUFormer, depending on the choice of $M$. The QK-first formulation (Softmax-$\mathbb{S}oLA$ and SphereUFormer) must materialize the key/value for the $M$ nearest neighbors, so the activation memory scales linearly with $M$. Linear-$\mathbb{S}oLA$ avoids this by directly aggregating the key/value outer product in $H$ (In Figure 2, right, our implementation does not construct neighbors' key/value, rather directly accumulates in $H$), so the corresponding activation memory is independent of $M$ (the computation still scales linearly with $M$). Refer to the discussion of Section S3 for more on the activation-memory tradeoff in both the QK-first (Softmax-$\mathbb{S}oLA$) and KV-first (Linear-$\mathbb{S}oLA$) cases.

