# OpenReview forum: "Spherical SO(3) Equivariant Local Attention"
_ICML.cc/2026/Conference — ICML 2026 regular_

### Official Review · Reviewer_SQW8 · 2026-03-10

**Soundness:** 3
**Presentation:** 3
**Significance:** 3
**Originality:** 3
**Overall Recommendation:** 5
**Confidence:** 2

**Summary:**

This paper addresses the pervasive SO(3) equivariance problem in spherical signal processing. It moves away from absolute positional encoding methods like Local Position Embedding (LPE) and innovatively proposes SoLA, a spherical SO(3) equivariant local attention mechanism.
The method encodes the relative position (great-circle distance) between tokens using native spherical unit direction vectors. Through theoretical design, the similarity is automatically decoupled into a content term and a geometric term, both of which are rotation-invariant, achieving rotational invariance in attention similarity.
The paper validates through extensive experiments that SoLA achieves full SO(3) equivariance in tasks such as 360° depth estimation and semantic segmentation, showing no significant performance degradation under rotated inputs.

**Compliance With Llm Reviewing Policy:**

Affirmed.

**Final Justification:**

The authors have provided thorough, clear, and complete responses to every issue raised in my review. Therefore, I tend to raise my rating to 5.

**Key Questions For Authors:**

Please refer to the points listed in the **Weaknesses** section, specifically regarding:

1.  Details of the gating mechanism.
2.  Transparency of experimental setup (data processing, environment) compared to SphereUFormer.
3.  Analysis of the increased FLOPs in Table 1.
4.  Clarification on the experiment in Section 4.3 and whether it tests equivariance or axis inference.
5.  Potential impact of interpolation distortion on rotated test results.

**Limitations:**

- **Gravity Prior Trade-off**: Full SO(3) equivariance, by design, prevents the model from leveraging absolute positional clues like the gravity direction. This could be a performance bottleneck in real-world scenarios where such priors are informative and consistent.
- **Grid Specificity**: The method is only validated on structured spherical grids (specifically, the icosahedron). For a paper on general spherical signal processing, adaptation or discussion regarding applicability to other spherical data formats (e.g., equirectangular, HEALPix) would strengthen the work.

**Strengths And Weaknesses:**

### A. Strengths

1.  **Theoretical**: The core idea of outer product modulation uses a unified operation to simultaneously convey content information and inject geometric information. Compared to SphereUFormer's LPE lookup table or some complex distance kernels, this design is more concise and straightforward to implement.
2.  **Novelty**: Modeling relative positions using sphere-native unit direction vectors and great-circle distances eliminates dependence on absolute grid coordinates. This significantly alleviates the computational efficiency issues of spherical signals and aligns perfectly with the geometric essence of spherical data, representing an innovation in positional encoding for spherical signal processing.
3.  **Experiments**: The multi-task design demonstrates the generality of SoLA. The core comparisons precisely isolate the performance gain attributable to the SoLA attention mechanism. The experimental design is logically rigorous and covers a comprehensive range of scenarios.

### B. Weaknesses

1.  **Theoretical**:
    - The gating coefficients appear to be a key module for "dynamically modulating relative position weights" in SoLA. However, specific implementation details are missing. Could the authors provide further elaboration?
2.  **Experimental**:
    - Data processing is not fully transparent. It should at least be explicitly stated that the data processing and experimental environment strictly follow those of SphereUFormer.
    - It is noted that in Table 1, the FLOPs for the SoLA method are approximately 30% higher compared to the SphereUFormer baseline. The reason for this significant increase in computation should be explained. For instance, the calculation in Equation (4) requires computing the inner product twice, not once. The computational load hasn't decreased; it seems shifted from memory consumption to computation.
    - In Section 4.3: "To enable the model to estimate the axis implicitly, we concatenate states at 15-minute intervals as additional input." This essentially informs the model about the rotation axis; the model could infer the axis by analyzing the trajectory rather than relying on equivariance for generalization. This makes it difficult to distinguish between the two possibilities based on the results in Table 2. Additional clarification or an ablation study is needed.
3.  **Rotation Tests**:
    - Generally, spherical data cannot be rotated directly. Rotation through interpolation methods might introduce some data distortion (potentially minimal). However, the rotated test results in Table 1 seem entirely unaffected by such potential distortion. Could the authors address this concern?

---

> ### Author Rebuttal · Authors · 2026-03-30
>
> Thank you for the detailed and constructive review.
> We especially appreciate the recognition that outer-product modulation offers a concise and novel way to couple content and geometry.
>
> ### Q1. Details of the gating mechanism
> Following the suggestion, we will elaborate more on the implementation details, including the following points.
>
> We compute the input-dependent gating coefficients $b^{Q}, b^K, w^Q, w^K$ in parallel with the QKV projection inside the same linear operation.
>
> #### KV-first case (for Linear-SoLA)
> We construct the $N\times 4$ modulation vectors and apply them to the query/key before performing the key-value aggregation (Eq. 3). We then aggregate the KV outer-product directly into $H$ using a custom Triton kernel.
>
> #### QK-first case (for Softmax-SoLA and Linear-SoLA)
> A naive implementation would construct the modulation matrix with Eq. 3, and then compute the attention map using the modulated vectors.
> We can bypass the 4D modulation for efficiency (Section 3.3).
>
> We first compute the query-key dot product of the unmodulated vectors, then construct the decay mask $M \in \mathbb{R}^{N \times M}$ using the input-dependent gating and normals as
> $$
> M_{ij} = [b_i^Q,w_i^Q \mathbf{u}_i^{\top}] \times [b_j^K,w_j^K \mathbf{u}_j^{\top}]^{\top},
> $$
> and apply that decay mask to the attention logits.
> The $M$ nearest neighbors of $b^K$ and $w^K$ are computed in the same way as the neighbor keys and values using the pre-computed indices.
>
>
> ### Q2. Experimental setup compared to SphereUFormer
> As the reviewer suggests, the comparison setup should be more explicit.
> We will clarify the following point in the revision.
> Our implementation is built on the SphereUFormer codebase.
> For depth estimation and the SS task, we use the same data processing, augmentation, training schedules, and evaluation protocols as SphereUFormer.
> We also adopt the same network architecture, an attention block structure including its feature dimensions.
>
> ### Q3. Analysis of the increased FLOPs in Table 1
> The main reason Softmax-SoLA has higher FLOPs than SphereUFormer is the input-dependent gating (Section 3.4).
> SoLA removes the position-dependent attention bias construction, reducing FLOPs and improving hardware friendliness.
> Still, gating adds more arithmetic than that saves, so the total FLOPs increase.
> This is why the FLOPs are higher while the measured latency remains comparable.
>
> ### Q4. Clarification on the SWE experiment
> Thank you for flagging this, and we apologize for the confusion caused by the lack of detail.
> In our SWE experiment, the training data are not rotated, whereas the test data are randomly rotated; this was stated only in the supplement.
>
> In this setting, a model that is not SO(3)-equivariant (sensitive to rotation axis) can exploit the fixed training orientation and thus predict future states *without using the 15-minute input*.
> By contrast, an SO(3)-equivariant model could not tell the fixed axis, so it must learn to use the 15-minute input to make predictions even in fixed orientation.
>
>
> ### Q5. Impact of interpolation distortion on rotated test results
> As the reviewer points out, rotating spherical data introduces interpolation distortion.
> We think that the similar score on the rotated data stems from how we generate the rotated data and the sufficiently high input resolution relative to the signals' spatial frequency.
>
> We construct rotated inputs by first rotating the associated unit normals ($H \times W \times 3$) of the input 2D image, then using the same implementation as SphereUFormer, we sample on an $N$-icospherical grid using the rotated normals.
> The sparse data ($[N,3]$) generated in this way affects less than rotating the spherical data in the 2D grid representation.
> We will clarify these points.
>
> To analyze this point further, we evaluate the equivariance error:
> $$
> f(x_{rot}) - f(x)_{rot}
> $$
> which is the difference between the outputs from the rotated inputs and the rotated outputs from the original inputs (the difference between the rows in Fig. 8).
>
> The equivariance error of SphereUFormer is about 43\%, while that of SoLA is about 2\%.
> The SoLA error is quite small, but not zero.
> It indicates the accuracy changes for each input (sometimes better, sometimes worse), but the average accuracy across random rotations is similar to that of the canonical input.
>
> ### Q6. Gravity prior and tradeoff
> This discussion is related to Q2 of `XQ7p`.
> Please refer to that response.
>
> ### Q7. Applicability to other spherical grids
> We expect the extension of SoLA to 2D parameterized spherical grids, such as equiangular and Legendre-Gauss grids, to be a promising future work.
> In those settings, Linear-SoLA could be combined with a low-rank spherical harmonic transform for spherical local averaging, potentially further improving efficiency by eliminating the sparse operation entirely.
> We consider these explorations as outside the scope of this paper.
> In the revision, we will clarify this as a limitation.

---

> > ### Author Rebuttal · Reviewer_SQW8 · 2026-04-03
> >
> > The authors have provided thorough, clear, and complete responses to every issue raised in my review, so I will raise my score.

---

> > > ### Author Response · Authors · 2026-04-03
> > >
> > > Thank you very much for your insightful review and for your acknowledgment. We will clarify the points you raised in the revised manuscript.

---

### Official Review · Reviewer_XQ7p · 2026-03-11

**Soundness:** 3
**Presentation:** 3
**Significance:** 3
**Originality:** 3
**Overall Recommendation:** 4
**Confidence:** 3

**Summary:**

This paper proposes a mechanism called SoLA (Spherical $SO(3)$-equivariant Local Attention), designed specifically for processing spherical signals. Most existing spherical vision Transformers rely on location-dependent local positional encodings (LPEs), which not only break full $SO(3)$ equivariance but also introduce additional memory and computational overhead. To address this issue, SoLA achieves an LPE-free local attention mechanism by modulating the query and key features with the outer product of each token's 4D unit direction vector. This modulation allows the query-key similarity computation to naturally decompose into content affinity and great-circle distance, remaining invariant under global $SO(3)$ rotations. Furthermore, the authors derive a softmax-free linear variant based on this, thereby avoiding per-query neighbor feature instantiation. Experiments integrating this into a U-shaped architecture (SoLA U-Net) demonstrate strong robustness under arbitrary 3D rotations and excellent computational efficiency on tasks including 360° depth estimation, semantic segmentation, spherical PDE forecasting, and spherical MNIST.

**Compliance With Llm Reviewing Policy:**

Affirmed.

**Key Questions For Authors:**

1. The paper mentions that the linear SoLA variant (Linear-SoLA) significantly reduces memory usage, but current experiments primarily evaluate this linear variant on MNIST classification and PDE forecasting. Why wasn't Linear-SoLA comprehensively tested on more computationally intensive dense prediction tasks (such as 360° depth estimation and semantic segmentation)?
2. In Table 1, SoLA slightly lags behind SphereUFormer on canonical (unrotated) inputs. Given that gravity is often a strong prior in practical applications (like indoor scenes), would the authors consider designing an architecture in the future that can adaptively fuse additional global coordinate cues while maintaining the $SO(3)$ equivariance of the underlying features?

**Limitations:**

Yes.

**Strengths And Weaknesses:**

- Soundness: The paper is technically very rigorous. The mathematical derivation for eliminating LPEs via direction vectors is logically consistent, successfully inducing an attention kernel that incorporates a distance term. The experimental evaluation is comprehensive, covering controlled corner detection tests, real-world vision tasks, and complex scientific forecasting tasks. The authors honestly and objectively point out the model's slight performance lag in standard (unrotated) views due to the lack of gravity cues.
- Presentation: The structure is clear and easy to follow. The motivation is well-articulated, and Figure 1 greatly helps readers intuitively grasp the core mechanism of outer-product modulation. The paper thoroughly reviews related work (projection methods, spherical CNNs, spherical Transformers) and accurately positions its own differences and advantages.
- Significance: It addresses a tangible pain point. Many existing spherical Transformers are only equivariant to yaw rotations, which limits their robustness under non-standard camera poses. Completely eliminating LPEs not only enhances theoretical elegance but also reduces latency overhead caused by irregular memory access, holding high practical value for applications in fields like robotics and meteorology.
- Originality: It provides a refreshing perspective. Using direction vectors for distance-preserving positional modulation is a highly clever design. By implicitly expressing geometric distance through outer-product lifting, the mechanism breaks free from explicit LPE dependencies in the form of lookup tables. This combination demonstrates strong originality in the field of spherical Transformers.

---

> ### Author Rebuttal · Authors · 2026-03-30
>
> Thank you for the careful and supportive review.
> We especially appreciate the positive assessment of the paper's technical rigor and the clarity of its motivation and positioning.
>
> ### Q1. Choice of Softmax-SoLA over Linear-SoLA
> We chose Softmax-SoLA for the depth and SS tasks because it is more efficient, in both FLOPs and activation memory, than the linear variant when $M=19$.
>
> In the KV-first formulation, Linear-SoLA avoids explicitly materializing neighbor key/value pairs, so its neighbor-memory cost is constant with respect to $M$.
> However, in the KV-first case, it cannot use the efficient technique that bypasses 4D lifting (Section 3.3), which applies only to the QK-first formulation.
> So, there is a tradeoff between the two approaches: QK-first avoids 4D lifting but requires neighbor materialization, whereas KV-first does the opposite.
>
> Following the SphereUFormer's settings, we set $C_{win}=2$ ($M = 19$).
> In this setting, the QK-first formulation, which avoids 4D lifting, is more efficient; the cost of 4D lifting hinders the advantage of avoiding neighbor materialization at this neighborhood size.
>
> The break-even point lies between $C_{win}=2$ ($M=19$) and $C_{win}=3$ ($M=43$).
> For $C_{win}\geq 3$, Linear-SoLA requires less activation memory because the gain from avoiding explicit materialization of key/value pairs outweighs the increase from the 4D lifting.
>
> However, at $M=43$, the accuracy of both SphereUFormer and Softmax-SoLA dropped in the tasks.
> For example, on the Stanford2D3D SS task, mIoU is 71.5 and 71.6 for SphereUFormer and Linear-SoLA, respectively.
> We therefore used $M=19$, which matches the default of SphereUFormer.
>
> In higher-resolution settings, $C_{win} \geq 3$ might yield better accuracy, but we have not tested this setting.
>
> At $C_{win}=2$, the accuracy of Softmax-SoLA and Linear-SoLA is very similar.
> In the Stanford2D3D SS task, the mIoU of Linear-SoLA is 72.1 (Softmax-SoLA is 72.2).
> We chose QK-first for efficiency and adopted softmax to compare with SphereUFormer.
>
> We will clarify this tradeoff in the revision and report the results of Linear-SoLA ($C_{win}=2,3$) and Softmax-SoLA ($C_{win}=3$) in the supplement.
>
> Please refer to `Ct7z` (Q4) for an additional discussion of the experiments in which the KV-first formulation benefits more.
>
> ### Q2. Gravity cue
> We agree that it can be a good prior in many practical applications.
>
> > Would the authors consider designing an architecture in the future that can adaptively fuse additional global cues?
>
> Yes, a slight modification to the embedding could achieve this, as we discuss next.
>
> In the revised manuscript, we will: (1) clearly position the current study so that this paper fucus on the strict SO(3)-equivariance; (2) present a formulation for possible extension of SoLA that utilizes gravity cues to motivate future work; and (3) state that evaluation of this variant is not included as a limitation.
> We will explore this in future work.
>
> #### Extension to incorporate gravity cues
> One way to improve accuracy is to use the minimum equivariance for the target applications.
> The full SO(3) equivariance might be too strict for some applications.
> For example, a chair is usually upright, and using this prior can help the model recognize the object if gravity is known.
> In this case, restricted SO(3)-equivariance--where the model remains equivariant under longitudinal/latitudinal movement but sensitive to local in-plane rotation--would be a good choice.
> An extension of SoLA that induces an additional angle-to-gravity term, $\beta$, would achieve this equivariance by the following modulation:
>
> $$
> [b, wu^{\top}+w_{g} u_g^{\top}+w_h u_h^{\top} ],
> $$
> where $u_g$ and $u_h$ are the local gravity and horizontal vector.
>
>
> When $w_g$ and $w_h$ are zero, this reduces to the current formulation (Eq.2), inducing the geodetic distance $\delta$, which is SO(3)-equivariant.
> When they are nonzero, the query key dot product induces a $\beta$ term of the form.
> $$
> A \cos \delta + B \sin \delta \cos \beta + C \sin \delta \sin \beta.
> $$
> This extension allows the model to exploit the gravity cue while retaining equivariance to longitudinal/latitudinal movement (restricted SO(3)-equivariance).
> By making $w_g$ and $w_h$ input-dependent, the model can adaptively fuse gravity cues when they are helpful, while still being able to fall back on strict SO(3)-equivariance when they are not.
>
> **Note 1.** *SphereUFormer* and the *+SLSA* variants in the ablation (Table 4) do not have this property.
> Those methods directly embed the global vertical position; they can use the gravity cue, but their functions are sensitive to latitudinal movement.
>
> **Note 2.** There *also* exist many applications that do not have a preferred axis, such as our SWE forecasting, medical imaging, etc.
> Moreover, even for gravity-aware applications, strict SO(3)-equivariance would be more efficient at capturing low-level visual cues, as these cues may appear in arbitrary orientations.

---

> > ### Author Rebuttal · Reviewer_XQ7p · 2026-04-03
> >
> > I appreciate the authors’ detailed response and the additional analyses provided in the rebuttal. The clarifications help improve the clarity of the work and address several of my questions.

---

> > > ### Author Response · Authors · 2026-04-03
> > >
> > > Thank you very much for your thoughtful review and for carefully reading our rebuttal. We will make the two raised points clearer in the revision.

---

### Official Review · Reviewer_Ct7z · 2026-03-12

**Soundness:** 4
**Presentation:** 3
**Significance:** 3
**Originality:** 3
**Overall Recommendation:** 4
**Confidence:** 2

**Summary:**

The paper introduces Spherical SO(3)-Equivariant Local Attention (SoLA), a novel local attention mechanism tailored for spherical signals.

The authors also propose a softmax-free linear variant of SoLA that computes attention via key-value aggregation, bypassing per-query neighbor materialization to improve memory efficiency.

The method is evaluated on multiple tasks, including 360-degree depth estimation, semantic segmentation, and partial differential equation forecasting, demonstrating computational parity with baselines while substantially improving zero-shot robustness to arbitrary SO(3) rotations.

**Compliance With Llm Reviewing Policy:**

Affirmed.

**Key Questions For Authors:**

See weakness.

**Limitations:**

See weakness.

**Strengths And Weaknesses:**

**Strengths**

**Soundness:** While I am not an expert in this specific subfield, the theoretical foundations of the paper appear to be mathematically sound. The authors provide clear and accessible proof sketches in Remark 1 and Remark 2 , which help illustrate how the proposed outer-product lifting ensures that the similarity score depends exclusively on feature affinity and geodesic distance.

**Experiments:** The experimental setup is well structured to support the paper's core claim of zero-shot rotational robustness. Experiments with rotated sets and discussion on performance trade-off between gains in canonical sets and rotational robustness with gravity cues are interesting.

**Weaknesses**

**Lack of Ablation:** The authors introduce an "input-dependent gating" mechanism to control the weight of the geodesic distance term dynamically without providing quantitative ablation study to isolate its effectiveness and efficiency.

**Hardware Optimization:** While the paper argues for computational efficiency by eliminating location-dependent LPEs , the Softmax-SoLA formulation fundamentally relies on retrieving M nearest samples on the sphere (e.g., M=19 ) to define irregular local geodesic neighborhoods. Furthermore, the pre-softmax similarity computation involves an explicit dot product of 4D direction vectors. This irregular memory access pattern and modified attention kernel make it highly restrictive, if not impossible, to apply modern hardware-level optimizations like FlashAttention, a practical system-level limitation the authors do not discuss.

**Omission of Memory (VRAM) Metrics:** The authors acknowledge that explicitly materializing per-query neighbors increases memory usage. Despite this, they do not report effective VRAM usage for their experiments comparing SoLA against baselines like Elite360D and SphereUFormer.
**Limited Evaluation of Linear-SoLA:** While Linear-SoLA is presented as a highly efficient variant that avoids per-query neighbor materialization , it is only evaluated on a relatively simple Spherical MNIST digit classification task. The core dense prediction tasks exclusively utilize Softmax-SoLA. The lack of a direct comparison between Softmax-SoLA and Linear-SoLA on these higher-resolution main tasks raises substantial concerns regarding Softmax-SoLA's true memory overhead and scalability in practical scenarios.

**Presentation**

Strengths: The manuscript is well-structured and easy to follow. The core mathmetical presentation and SOLA architecture is effectively visualized.

Weaknesses: Overall figures (Figs. 3, 4, 5, 6, 7, and 8) consistently contains unnecessary gray lines.

---

> ### Author Rebuttal · Authors · 2026-03-30
>
> Thank you for the careful and constructive review.
> We especially appreciate the recognition that the outer-product lifting is mathematically well motivated and the positive assessment of the rotated-set experiments.
> We also appreciate the important suggestion for ablation.
>
> ### Q1. Ablation of the input-dependent gating mechanism
> Thanks for pointing out the important ablation for isolating the contribution of the input-dependent gating.
> Following the suggestion, we replaced the input-dependent gating with a learnable scalar per head (initialized all the ones).
> In the Stanford2D3D SS task, mIoU drops from 71.9\% to 70.8\%, while latency improved from 94.2 to 92.5 ms (FLOPs reduce 23\%).
> The decrease in latency is smaller than the FLOP reduction; we attribute this to the fact that we computed the gating coefficient in parallel with the QKV projection (L270).
> We will add this ablation, along with the corresponding image classification results, to Table 4 in the revised paper.
>
> ### Q2. Hardware optimization
> As the reviewer pointed out, both our Softmax-SoLA and SphereUFormer still need to gather $M$ nearest neighbors, which remains a bottleneck and prevents higher GPU throughput.
>
> The difference lies in the position-embedding part.
> SphereUFormer constructs an attention bias via location-dependent bilinear sampling from a learnable table (location-dependent LPEs), whereas SoLA constructs a decay mask from normals, thereby avoiding sparse bilinear sampling.
> In the QK-first formulation, the 4D lifting of query/key can be avoided for efficiency by constructing a *decay mask* (Section 3.3).
> Please also refer to Q1 from `SQW8` for the decay mask using input-dependent gating.
>
> SoLA's approach is more GPU-friendly in the position embedding part, but neighbor gathering for key-value remains the major bottleneck.
> We agree with the reviewer that the *gather* operation makes further hardware optimization difficult, if not impossible.
> We consider switching from the sampled point ($[N,C]$) to the 2D parameterized grid ($[H,W,C]$) a promising approach (Section 5, Q7 from `SQW8`), but we consider this outside the current study.
> We will clarify the above point.
>
>
> ### Q3. Memory metrics
> Thanks to the question, we have realized that we should make it clear that there are two distinct places where per query neighbors are considered: (1) attention bias and (2) key/value, each of which also needs a separate discussion for the QK-first (Softmax-SoLA) and KV-first (Linear-SoLA) cases.
>
> #### (1) Attention bias
> LPE-based methods (e.g., SphereUFormer) compute attention bias using local neighbors, which requires sparse, position-dependent, bilinear sampling.
> Both SoLA variants avoid this neighbor operation by novel position modulation.
>
> #### (2) Key-values
> As the reviewers point out, the QK-first formulation (Softmax-SoLA and SphereUFormer) must materialize the key/value for the $M$ nearest neighbors, so the activation memory scales linearly with $M$.
> Linear-SoLA avoids this by directly aggregating the key/value outer product in $H$ (In Fig. 2, right, our implementation does not construct neighbors' key/value, rather directly accumulates in $H$), so the corresponding activation memory is independent of $M$ (the computation still scales linearly with $M$).
>
> We will revise to make these distinctions explicit and report the actual VRAM usage for different $M$ in both variants.
>
> ### Q4. Choice of Softmax-SoLA over Linear-SoLA in depth and semantic segmentation
> This question closely relates to Q1 from `XQ7p`.
> Please refer to that response for details.
>
> #### Linear-SoLA with larger $M$ (extreme case)
> As discussed in Q1 from `XQ7p`, there is a break-even point in the choice of QK-first and KV-first for $M$ (in between $C_{win}=2$ ($M=19$) and $C_{win}=3$ ($M=43$)) in terms of activation memory and FLOPs.
> The extreme case is $M=N$, which corresponds to full attention.
> In this setting, the difference is significant: the activation memory of the QK-first formulation (Softmax-SoLA, or SphereUFormer) increases with $\mathcal{O}(N^2)$, easily exceeding the GPU's memory limits, while in the KV-first formulation, it is $\mathcal{O}(N)$.
> The image classification experiments in Section 4.4 demonstrate the effectiveness of Linear-SoLA in this extreme setting.
> In Table 3, we report results using a low-resolution input of icosphere rank 4 (2.5K points), roughly reflecting the resolution of the original MNIST data (28x28).
> We can increase this to rank 7 (328K points, same settings as the depths and SS task in Table 1) or even higher.
> We conducted the experiments with input rank 7 using Linear-SoLA.
> The accuracies were 96.5 and 94.2 for the canonical and rotated inputs (no significant improvement).
> We will also clarify this point.
>
> ### Q5. Unnecessary guidelines in Figs. 3–8
> Thank you for pointing this out, and we apologize for the issue.
> We have confirmed the problem in some PDF readers.
> We will ensure that these lines do not appear in the revision.

---

> > ### Author Rebuttal · Reviewer_Ct7z · 2026-04-05
> >
> > Thanks for the rebuttal. Most of my concerns are resolved. For now, i will hold my score, however also leans towards to raise the score after the discussion with other reviewers.

---

> > > ### Author Response · Authors · 2026-04-05
> > >
> > > Thank you very much for the acknowledgment. We also appreciate your constructive review, which highlights important aspects that will be clarified in the revision.

---

### Official Review · Reviewer_K9UJ · 2026-03-13

**Soundness:** 4
**Presentation:** 3
**Significance:** 4
**Originality:** 4
**Overall Recommendation:** 5
**Confidence:** 4

**Summary:**

This paper develops a new attention mechanism which incorporates spherical invariance data for signals which are observed on a sphere without the use of local positioning embeddings. Other approaches to attention mechanisms for spherical data require the use of local positional embeddings, which can be computationally expensive and may not capture the full rotational invariance of the data. Here, authors show that incorporating rotational invariance aids in the reconstruction of spherical signals, and that this new mechanism is robust to rotational transformations of the data. Authors' method, SoLA, is based on taking key-query pairs for features located on a sphere and lifting them to a higher dimension in order to incorporate angle/distance information. Each point $i$ has a query feature vector $q_i$ and a unit direction vector $u_i$, while $j$ has key feature vector $k_j$ and unit vector $u_j$. We want local attention on the sphere, so our score should be something like $s_{ij} = \langle q_i, k_j \rangle g(\cos(\delta_{ij}))$ where $\cos(\delta_{ij}) = u_i^Tu_j$.Simplest choice of $g$ is affine: $g(\cos(\delta_{ij})) = b + w\cos(\delta_{ij})$. Trick is that $s_{ij} = \langle q_i, k_j \rangle  (b + w\cos(\delta_{ij}))$ can be written as the Frobenius inner product of outer products: $\langle a \otimes c, b \otimes d \rangle_F = \langle a, b \rangle \cdot \langle c, d \rangle$. Therefore $s_{ij} = \langle q_i, k_j \rangle  (b + w\cos(\delta_{ij}))$ can be written as $s_{ij} = \langle q_i \otimes [b, wu_i^T], k_j \otimes [1, u_j]\rangle_F$ where $[b, wu_i^T], [1, u_j]$ are concatenated 4 vectors. This factors as (content) times (geometry), meaning we can use linear attention, which saves a bunch of memory and FLOPs. Main comparison from related works isbe SphereUFormer, which is about as good as SoLA at 360 degree image reconstruction. For arbitrary rotations, SoLA blows SphereUFormer out of the water, though SoLA takes a lot more FLOPs --- authors point out that while positional embeddings used in comparison model are low FLOP, they are time costly. On other data, SoLA is way better than SphereUFormer, including shallow water simulations which SphereUFormer also ran.

**Compliance With Llm Reviewing Policy:**

Affirmed.

**Final Justification:**

No major updates during the rebuttal process to my review. I quite like this paper and score will remain where it is.

**Key Questions For Authors:**

- Can you attempt rewriting Section 3.2 to better describe ideas including "modulat[ing] each query and key by lifting their features using the outer product of four-dimensional global position vectors"?
- Is SoLa more or less memory efficient than SphereUFormer? I get the sense it's not more memory efficient, but it sort of doesn't matter. Can you unify this treatment?

**Limitations:**

Yes

**Strengths And Weaknesses:**

Strengths:
- Excellently written and presented paper on a critical topic that achieves a novel advance and shows improvement over related models in all experiments
- SoLA does not need positional embeddings on the sphere, which unlocks more 360-degree data and also takes advantage of natural symmetries in the modality
- Discussion of spherical MNIST dataset is highly intriguing and generalizes this approach for audiences outside of spherical data experts

Weaknesses:
- Section 3.2 is difficult to follow
- The discussion as to SoLA's relative memory performance is muddled

---

> ### Author Rebuttal · Authors · 2026-03-30
>
> Thank you for the thoughtful and encouraging review. We especially appreciate the concise summary of the core mechanism and the positive assessment of the paper's novelty and significance.
>
> ### Q1. Clarifying the modulation mechanism in Section 3.2
> We agree that Section 3.2 can be made much clearer so that the reader can grasp the key idea intuitively before moving to the algebraic details and proofs.
> In the revision, we will first present the intuitive mechanism by further elaborating on it using the illustration in Fig. 1, while taking the reviewer's concise summary into account.
> A 4D direction-dependent vector modulates each query and key, and the resulting similarity decomposes into a content term and a geometry term based on the great-circle distance.
> We will then move to the proof of the induced similarity and SO(3)-equivariance after this intuitive explanation.
>
> ### Q2. Memory efficiency of SoLA
> Thanks to the reviewer's suggestions, we find that the current presentation mixes different notions of "memory efficiency".
> The reviewer's understanding is right on this point; for the Softmax-SoLA setting used in the dense prediction experiments (depth and SS), SphereUFormer consumes less activation-memory in our settings (i.e., same QKV feature dimension), mainly because SoLA adds input-dependent gating (Section 3.4) and a following decay mask computation (please also refer to Q1 of `SQW8` for the implementation details of the input-dependent gating).
>
> The advantage of SoLA is two-fold:
> (1) It removes the position-dependent sparse bilinear sampling used to build local positional embedding biases, thereby avoiding irregular memory access and improving hardware friendliness.
> (2) In addition, Linear-SoLA (KV-first) avoids explicit per-query neighbor materialization, so the memory required for neighbor key/value features no longer grows with $M$.
> In this case, the activation memory of SoLA would be smaller than that of SphereUFormer, depending on the choice of $M$.
> Please also refer to the discussion of Q1 in `XQ7p` for more on the activation-memory tradeoff in both the QK-first (Softmax-SoLA) and KV-first (Linear-SoLA) cases.
>
> In the revision, we will explicitly separate these points for clarification: increased activation memory for the softmax variant, hardware efficiency from removing LPE lookups of both variants, and the $M$-independent neighbor-memory (feature-map) benefit of KV-first (Linear-SoLA) formulation.

---

> > ### Author Rebuttal · Reviewer_K9UJ · 2026-03-31
> >
> > I find the authors' rebuttal especially as regards further discussion of the memory efficiency (concerns which were shared by other reviewers) to be sufficient for me! My concerns are addressed, score remains a 5.

---

> > > ### Author Response · Authors · 2026-04-02
> > >
> > > Thank you very much for your positive assessment and for the thoughtful review.  We will make these points clearer in the revision.

---

### Decision · Program_Chairs · 2026-04-30

**Decision:**

Accept (regular)

**Comment:**

Initial reviews were unanimously positive, with reviewers highlighting the solid theory, novelty, great presentation and strong experiments. Only minor concerns were raised about a missing ablation, memory and compute efficiency.

The rebuttal clarified all reviewers concerns and one of them increased their already positive score.

I follow the reviewer consensus and recommend acceptance.